# Monitoring high-intensity focused ultrasound thermal therapy by ultrasound doppler imaging using twinkling artifact

Amirhossein Jamallivani[1], Hamid Behnam[1]*, Jahangir Tavakkoli[2,3]

1 Department of Biomedical Engineering, Iran University of Science and Technology, Tehran, Iran, 2 Department of Physics, Toronto Metropolitan University, Toronto, ON, Canada, 3 Institute for Biomedical Engineering, Science and Technology (iBEST), Keenan Research Centre for Biomedical Sciences, St. Michael's Hospital, Toronto, ON, Canada

* behnam@iust.ac.ir

## Abstract

High-Intensity Focused Ultrasound (HIFU) is a non-invasive therapeutic modality that uses high-energy acoustic waves to thermally coagulate tissue at the focal region. The Twinkling Artifact (TA) is a color Doppler artifact caused by the acoustic radiation force and the consequent tissue vibration during Doppler imaging. This paper aims to employ TA for real-time detection and monitoring of HIFU-induced lesions. A dataset gathered in a previous study concerning ex vivo porcine tissue samples was used, in which the real-time backscattered radiofrequency signals were acquired before, during, and after HIFU treatment. To investigate the presence of TA in Doppler images, the amplitude of each pixel is considered in the sequence of frames as time-series or slow-time signals. It is shown that the main frequency of slow-time signals represents the Doppler frequency shift. Doppler images were constructed using the maximum frequency from every 10-sample slow-time signal. By constructing Doppler images, the frequency shifts within tissue during HIFU treatment were visually and analytically assessed. Our frequency analysis of RF data confirmed the occurrence of TA during HIFU exposure. Furthermore, a novel method was developed for lesion formation monitoring, with less than a 5% error rate in depth and width measurements for depicting coagulated tissue dimensions.

## 1. Introduction

In recent years, the development of High-Intensity Focused Ultrasound (HIFU) has inspired great hopes in revolutionizing non-invasive medical treatments. HIFU uses a focused transducer to emit high-energy acoustic waves to heat and coagulate tissue within a focal region. HIFU has demonstrated promising research trials in therapeutic applications and has even found some clinical applications. To broaden the applications of HIFU in clinical procedures, developing real-time methods for detecting and monitoring thermal lesions is essential [1].

**Data availability statement:** All the radio frequency (rf) data files used in this study, along with the two MATLAB codes developed for reading them, are publicly available in a Zenodo repository at https://doi.org/10.5281/zenodo.15188602.

**Funding:** The author(s) received no specific funding for this work.

**Competing interests:** The authors have declared that no competing interests exist.

Despite its potential, a significant challenge persists. The primary difficulty involves devising a detection and monitoring technique that not only detects coagulated tissue accurately but also provides quantitative distinctions between the coagulated tissue and its surrounding unexposed tissue in real-time during treatment. Such capabilities are critical to ensure the safety and efficacy of HIFU procedures, as noted in previous studies [2].

In the last two decades, several studies have focused on HIFU thermal lesion detection, including Magnetic Resonance Imaging (MRI) [3–6] and ultrasound imaging [7]. Among these, ultrasound B-mode imaging is the simplest method for HIFU lesion detection [8]. Compared to MRI-guided methods, ultrasound imaging offers advantages such as lower cost, portability, compatibility with HIFU exposure, and a simple therapy setup. HIFU lesions are indicated by an increase in brightness in B-mode ultrasound images [8].

Although ultrasound-based methods have advantages over their MRI-based counterparts, employing ultrasound for temperature measurements faces significant challenges due to nonlinear responses [9,10], thermal expansion, variations in attenuation and absorption [11–13], and different sound speeds across tissue types [14]. The HIFU-induced thermal lesions were effectively detected by Nakagami parametric imaging, based on the distribution of ultrasound backscattered signals [15].

The time series analysis of RF signals has demonstrated the potential to assess HIFU lesion formation in tissues [16,17]. In addition to these techniques, considerable effort has been directed toward developing alternative ultrasound-based methods for monitoring HIFU treatments [18–20]. These include ultrasound elastography [21–23], vibroacoustography [24], and local harmonic imaging (LHI) [25,26].

Ultrasound elastography produces elastograms that illustrate tissue responses to strain or stress. In a study by Arnal et al. [27], HIFU-exposed tissue shear wave elastograms were reported every three seconds in both in vitro and in vivo animal subjects. This article reported elasticity changes of up to 30% in liver tissue and 400% in muscle tissue. Bing et al. [28] used Acoustic Radiation Force Impulse (ARFI) imaging to monitor HIFU thermal ablation. They reported an up to 54% error in lesion area estimation. Thierman et al.[24] tried to monitor HIFU ablation by vibroacoustography. This technique requires the use of two ultrasound transducers to induce controlled vibrations within the targeted tissue. These vibrations are then measured with a hydrophone. Vibroacoustography is limited by factors such as the number of transducers required and their optimal spatial arrangement for generating and monitoring vibrations within the target region.

Local Harmonic Imaging (LHI) [29], another advanced technique, uses ultrasound to detect HIFU-induced thermal lesions by applying an acoustic radiation force that produces Localized Harmonic Oscillations (LHO). High frame rate ultrasonography is needed to measure changes in LHO amplitude. The experiments on HIFU exposed ex vivo porcine muscle showed an increase in Young's moduli and a decrease in LHO amplitude.

Recent advancements in ultrasound-based HIFU lesion monitoring have demonstrated significant progress in improving treatment precision and safety. Liu et al. [30]

validated the feasibility of Acoustic Radiation Force Impulse (ARFI) imaging for monitoring HIFU thermal damage, highlighting its ability to provide clearer lesion boundaries and higher damage contrast compared to conventional methods. Yang et al. [31] proposed multiple ultrasonic parametric imaging techniques, identifying horizontally normalized Shannon entropy (hNSE) imaging as the most effective for lesion recognition due to its superior contrast resolution and real-time monitoring capabilities. Additionally, Zhou et al. [32] introduced weighted ultrasound entropy (WUE) imaging, which showed a 39.2% to 53.4% improvement in contrast-to-noise ratio (CNR) over B-mode imaging, demonstrating enhanced sensitivity and accuracy in HIFU ablation monitoring. These studies underscore the importance of developing advanced ultrasound techniques for HIFU therapy, as they address critical challenges in real-time lesion assessment and pave the way for more reliable clinical applications.

Twinkling Artifact (TA) is a color Doppler imaging artifact that was discovered accidentally by radiologists. TA appears as some pixels that rapidly change color from red to blue and from blue to red, creating a twinkling effect. The initial report on TA by Rahmouni et al. in 1996 [33] marked the discovery that this artifact does not originate from blood flow or movement [33–39]. This led to multiple theories to explain the cause of TA, where coloration is not exclusively linked to flow or major movements. It has been observed adjacent to regions of pronounced vascular stenosis [40] or even within echo-free regions devoid of flow [35,41].

Numerous studies have posited that TA correlates with textural irregularities of the imaged medium under sonographic examination [33–37,42]. Since its accidental discovery, a substantial body of research has been dedicated to deciphering the fundamental mechanisms responsible for the generation of TA [37–39,43,44]. There are also efforts to use this artifact for diagnostic purposes; for instance, researchers have investigated the ability of TA to detect breast microcalcifications [45,46]. In several studies conducted on the twinkling artifact by Jamzad et. al.[42,47–49], they worked on different factors of TA appearance in Doppler Images. They also worked on classifying TA in Doppler images acquired from sandpapers with seven levels of roughness. They successfully developed a 7-class classifier with 98.33% average accuracy. These studies showed that TA can effectively recognize the roughness of phantoms. However, this result is highly dependent on the ultrasound imaging instruments and cannot be easily extended to other instruments and projects. However, these studies investigated important statistical features in TA detection and classification.

Recent studies have demonstrated the potential of ultrasound-based techniques for detecting microbubbles and mapping cavitation activity in HIFU therapy. For instance, Khokhlova et al. [50] explored the use of the Twinkling Artifact (TA) to monitor cavitation microbubbles, while Tong Li et al. [51] proposed the 'bubble Doppler' technique for active cavitation mapping. These studies have shown promising results in enhancing our understanding of cavitation dynamics during HIFU treatments. However, a research gap remains in the application of TA for thermal lesion monitoring in HIFU therapy. Addressing this gap, our study proposes a novel application of TA to investigate its presence and characteristics in the lesion area, offering a new approach for real-time assessment of HIFU-induced thermal lesions. This work aims to complement existing methods by providing a cost-effective and accessible tool for improving treatment precision and safety.

This article investigates the presence of the Twinkling Artifact in the monitoring of HIFU-induced thermal lesions in ex vivo tissue. The core concept is based on the hypothesis that vibration induced by Acoustic Radiation Force (ARF) in coagulated tissue can cause TA. Since HIFU-induced thermal lesions demonstrate distinct acoustic and mechanical properties compared to native tissue, we expect corresponding differences in frequency-domain behavior at the lesion site. Consequently, our study aims to: (1) quantify frequency-domain disparities through observed Doppler-type shifts, and (2) evaluate Twinkling Artifact presence in Doppler reconstructions.

Our dataset in this research was composed of radio frequency signals backscattered from HIFU-exposed tissue. This data allowed us to investigate statistical and frequency features, including the twinkling artifact. After showing that ultrasound Doppler imaging could be beneficial in US-guided HIFU therapy, more investigation can take place with commercial Doppler imaging systems. The subsequent section describes the dataset employed and delineates the methodology for Doppler signal extraction as well as the construction of Doppler images.

## 2. Method and materials

### 2.1. Data

In this study, we apply our proposed methods to ultrasound RF echo data which was acquired previously at Toronto Metropolitan University (formerly Ryerson) [19]. Fresh ex-vivo porcine muscle tissues were utilized in the experiments. In the data acquisition phase, researchers used a confocal arrangement of a single-element HIFU transducer and a 192-element endocavity array probe imaging system. The HIFU transducer (Model 6699 A101; Imasonic, Voraysur1'Ognon, France) was spherically concave with a center frequency of 1 MHz, F-number of 0.8 and aperture diameter of 125 mm. For recording B-mode images and RF signals, the ultrasound imaging system (Sonix RP, Ultrasonix, Richmond, BC, Canada) and a convex array probe (EC9–5/10, Ultrasonix) with 192 elements with a center frequency of 6.5 MHz and bandwidth of 3 MHz were used. More details on experimental setup are available in the referenced paper. The schematic diagram of the experimental setup is depicted in Fig 1.

In the experiments, RF signals and B-mode frames were obtained before, during, immediately after, and 10 minutes after HIFU exposure at various total acoustic powers of 90, 110 and 130 W. These power levels were selected because coagulative necrosis, a key indicator of thermal lesions, typically occurs at higher acoustic powers. This approach aligns with the aim of our project, which is to investigate the presence of the Twinkling Artifact (TA) in HIFU-induced lesions. The data has been segmented into three parts: Pre-HIFU, Dur-HIFU, and Post-HIFU. For Pre-HIFU, RF data acquisition targeted normal tissues. During the Dur-HIFU, RF data were collected while the HIFU transducer was momentarily switched off for 120 ms during exposure to avoid possible interference between HIFU and imaging transducer. Finally, Post-HIFU data were gathered 10 minutes following the HIFU sonification. Fig 2 illustrates the specific timing of HIFU exposure and the sequence of collected signals including Pre-HIFU, Dur-HIFU, and Post-HIFU. For more technical information about the transducer and experimental setup refer to the [19].

For each acoustic power setting, the dataset includes 81 frames, divided into 16 Pre-HIFU frames, 49 frames during and immediately after HIFU exposure (40 during HIFU and 9 immediately post-HIFU), and 16 frames acquired 10 minutes post-HIFU. Dimensional information of each frame of acquired data is presented in Table 1.

### 2.2. Methodology

In this study, we investigate the presence of the Twinkling Artifact (TA) in thermal lesions induced by High-Intensity Focused Ultrasound (HIFU). The dataset used in this work consists of backscattered RF signals, as Doppler signals

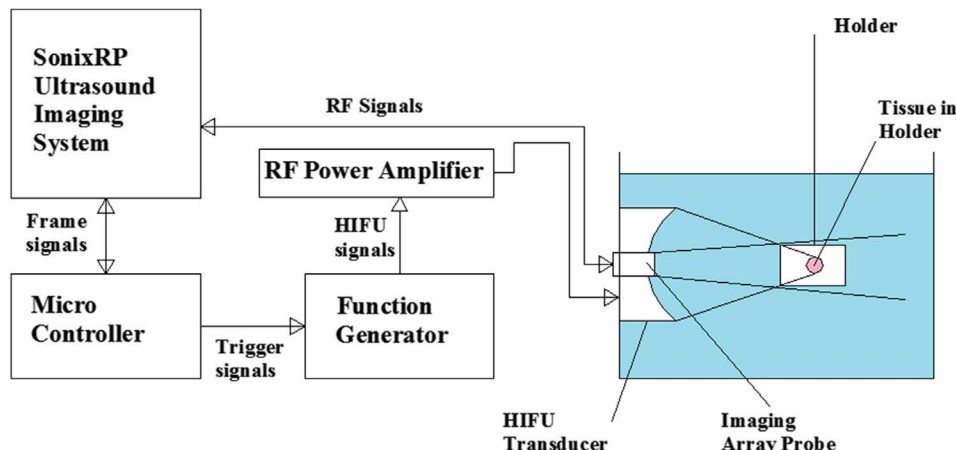

**Fig 1. Schematic diagram of the experimental setup for image-guided High-Intensity Focused Ultrasound (HIFU) [19].**

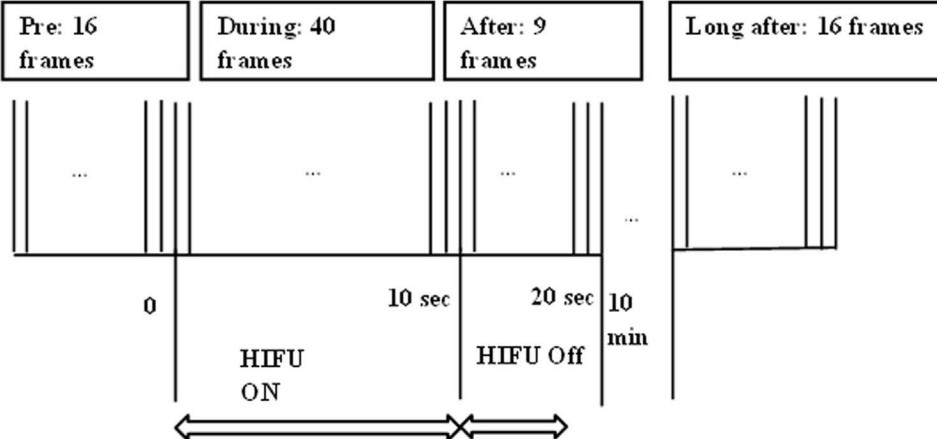

**Fig 2. Timing sequence of High-Intensity Focused Ultrasound (HIFU) exposure and radiofrequency echo data acquisition [19].**

**Table 1. Dimensional Information of Each Frame of Acquired Data (Number of RF Lines, Samples per Line, and Dimensions in mm).**

|  | Samples per line (representing depth in mm) | Lines (representing width in mm) |
|---|---|---|
| Original | 4680 (90.10 mm) | 192 (64.00 mm) |
| Selected Region (contain tissue and lesion) | 1000 (19.25 mm) | 40 (13.33 mm) |

were not directly acquired. Therefore, the first step involved extracting Doppler signals from the RF data, followed by the construction of Doppler images. The proposed method is capable of processing signals containing all samples in each time series. However, to enable real-time reporting and effectively guide HIFU treatment, we opted to calculate frequency features from segmented time signals. We tested segment sizes ranging from 1 to 10 samples and ultimately selected 10-sample segments with a one-sample step size and nine-sample overlap. This approach balances computational efficiency with sufficient temporal resolution.

Fast Fourier Transform (FFT) was applied to these segmented signals to obtain the phase and amplitude of each segment. The maximum phase of each signal was extracted, and both the phase and corresponding amplitude were stored for each pixel. To extract frequency from the phases, we first removed noise effects by applying an amplitude threshold, set at one-tenth of the maximum amplitude. This threshold ratio was determined through iterative testing to optimize noise reduction while preserving meaningful signal components. Frequencies were then calculated by subtracting the phase of each pixel in two consecutive records. These frequencies represent the Doppler shift frequency, and by visualizing the matrices of these extracted frequencies, we generated the Doppler images. In the following sections, we provide a detailed explanation of the methodology used in this study, including the theoretical foundations and practical implementation of the proposed approach.

**2.2.1. Doppler signals.** In this manuscript, we report a real-time method that can be integrated with existing ultrasound devices in medical centers. We investigated the potential of Doppler imaging for monitoring lesions formed by High-Intensity Focused Ultrasound (HIFU) ablation. To tailor this approach, we developed a method for extracting the Doppler shift from backscattered radiofrequency (RF) signals. This method is based on the velocity estimation methodologies described in Foundations of Biomedical Ultrasound by Cobbold [52] (Section 10.5 physical principles of pulsed systems). According to the following descriptions and analysis provided, the main frequencies of sampled signals from RF signals qualify as Doppler shift frequencies. First, the Doppler frequency shift can be expressed as follows (1):

$$f_D = \frac{2v}{c_o} f_0 \, \cos\theta \tag{1}$$

In this equation(1), $f_D$ stands for the Doppler shift frequency, $f_0$ demonstrates the transmitted frequency, $c_0$ is the wave velocity in the medium, $\theta$ is the angle between the direction of wave propagation and the direction of motion and $v$ is the velocity of the target. This formula is derived by subtracting the transmitted signal frequency from the received signal frequency (2.a):

$$f_D = f_R - f_T = f_0 \left[\frac{2v}{c-v}\right] \tag{2.a}$$

In this equation (2.a) $f_T$ is the transmitted frequency and $f_R$ is the received frequency from a moving scatterer that is calculated from this equation (2.b):

$$f_R = f_0 \left[\frac{c+v}{c-v}\right] \tag{2.b}$$

In Biomedical Ultrasound, Cobbold derives an important equation for estimating the velocity in a pulsed system from a single scatterer with constant velocity. This approach can provide an alternative method for calculating the Doppler frequency shift. If we denote the velocity of the scatterer as $v$ and the pulse repetition interval as $t_{PRI}$, the displacement along the beam axis during successive transmissions is calculated as $\Delta z = (v \, cos\theta)t_{PRI}$. Consequently, the corresponding change in the delay between sequentially received signals is $\Delta \tau = \frac{2\Delta z}{c_0}$, which can be further reformulated as (3.a):

$$\Delta \tau = \frac{2(v \, cos\theta)t_{PRI}}{c_0} \tag{3.a}$$

The velocity of the scatterer can be calculated by the following equation (3.b):

$$v = \frac{c_0 \Delta \tau}{2 \, t_{PRI} \cos\theta} \tag{3.b}$$

This equation is derived from equation (3.a) by rearranging the terms to solve for $v$. RF signals in this context are called fast-time signals. By sampling from a specific depth in a sequence of received RF lines, we obtain a signal known as a slow-time signal. The central frequency of the slow-time signal is approximated by $2\pi f \approx \frac{\Delta \phi}{t_{PRI}}$, where $\Delta \phi$ is the phase shift of the slow-time signal. However, the phase shifts of the slow-time and fast-time signals are identical, so that $\Delta \phi = 2\pi f_c \Delta \tau$, where $f_c$ is the center frequency of the transmitted waveform. Consequently, the center frequency of the pulse wave, or slow-time signal, can be re-expressed as $f = \frac{f_c \Delta \tau}{t_{PRI}}$. After substituting $\Delta \tau$ as given by Eq.3.a, the important equation can be derived as follows:

$$f = \frac{2v \cos\theta}{c_0} f_c \tag{4}$$

Equation 4 matches the frequency shift formula found in Equation 1. Thus, we can use the slow-time signal's frequency as the RF signal's frequency shift. From the RF data, we extract the frequency characteristics of the slow-time signal for each pixel in the sample tissue. Doppler images are then a visual representation of these frequency characteristics. (You can refer to the 'Fundamentals of Biomedical Ultrasound' by Cobbold [52] for further information about deriving equations).

**2.2.2. Doppler images.** In this phase, the Doppler signal was calculated for all samples. During the 10-second HIFU treatment, 40 frames were captured at 250 millisecond intervals. To obtain the Doppler images, we applied the Fast Fourier Transform (FFT) to slow-time signals. We had slow time signals with 40 samples during HIFU exposure. First, frequency features were extracted from non-segmented signals, so we only had one picture during HIFU exposure. To improve the temporal visualization, every 10 samples of the slow-time signal were considered as a segment of the slow-time signal. By sliding windows across slow-time signal with 1 sample step, we extracted 30 windows from HIFU exposure. After extracting the amplitude and frequency components from every 10-sample segment of the slow-time signals, the noise effects in the amplitude are omitted, and the maximum frequency is designated as the frequency shift for that pixel.

## 3. Result

### 3.1. Slow-time signals

In this paper, the RF signals are called fast-time signals due to their high frequency. Also, the slow-time signals are sampled signals from a specific depth in a sequence of RF lines. The time interval of slow-time signals depends on signal acquisition. In this case, the rate is 40 signals in 10 seconds during HIFU exposure. The slow-time signals for all frames in different powers were extracted. The RF (fast-time) and slow-time signals from the 110th RF line (which includes the lesion formation data) are shown in Fig 3. The RF line number 110 is selected because we already know from B-mode images that the lesion is formed in the regions between 90–130 and depth between sample 3000–4500.

It can be seen that with HIFU exposure, there are changes in the amplitude of signals. In the next stage, we applied FFT to each slow-time signal and extracted the maximum frequency and amplitude of each sample in all RF lines in the selected depth; for instance, the 110th RF line's Maximum amplitude, frequency and filtered maximum frequency are illustrated in Fig 4.

In the Methodology section, we showed that the main frequency of slow-time signals can be represented as Doppler frequency shifts. Therefore, we applied FFT to extract the maximum frequency and amplitude of each slow-time signal. Although there are closely spaced frequencies at each depth, we can disregard frequencies with low amplitudes by applying a threshold. In Fig 4c, the maximum frequencies of considerable amplitudes of slow-time signals are shown.

To construct real-time Doppler images from our data, we consider 10-length windows from slow-time signals, advancing one step forward across time (or frame numbers). Then, the process of extracting the maximum frequency, which is explained here for the 110th RF line, was applied to selected region signals. First, all slow-time signals were saved in

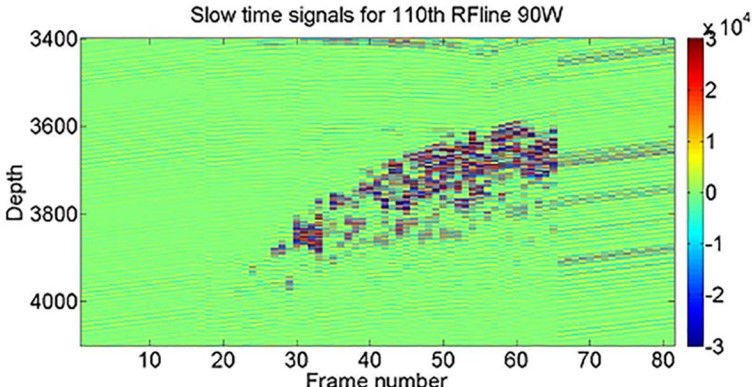

**Fig 3. This image illustrates the 110th RF line in 81 records of 90W HIFU exposure.** Each vertical line represents a selected depth of RF signals (sample 3400 to 4100) or a fast-time signal, while each horizontal line represents a slow-time signal. (The colorbar unit is the amplitude of the received RF signal).

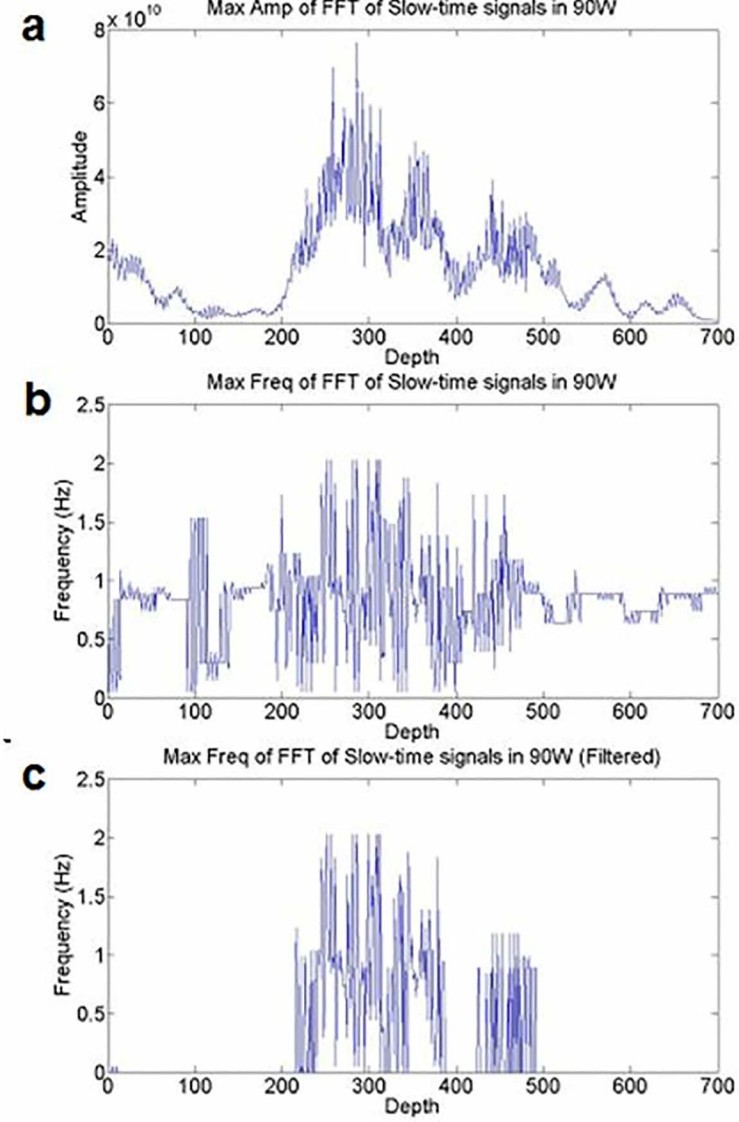

**Fig 4.  a) Maximum amplitude, b) frequency, and c) filtered maximum frequency of slow-time signals from the 110th RF line, samples 3400 to 4100, under 90W HIFU exposure.**

tensors. Then, FFT was applied to each 10-sample window, and the considerable maximum frequencies were saved. Finally, Doppler images were constructed for every 10 frames. In the next section, the constructed images are shown.

### 3.2.  Doppler images

To achieve real-time lesion monitoring, Doppler images were constructed by windowing the slow-time signal. A selection of these images, which illustrate lesion formation under 90W HIFU exposure, are shown in Fig 5.

In Fig 5, we illustrate some frames of the constructed images that show HIFU ablation step by step in the sample. The 13th, 26th, and 39th frames correspond to when HIFU was active, and the 67th frame is from 10 minutes after turning off HIFU. Therefore, the images constructed from slow-time signals enable the visual monitoring of lesion formation.

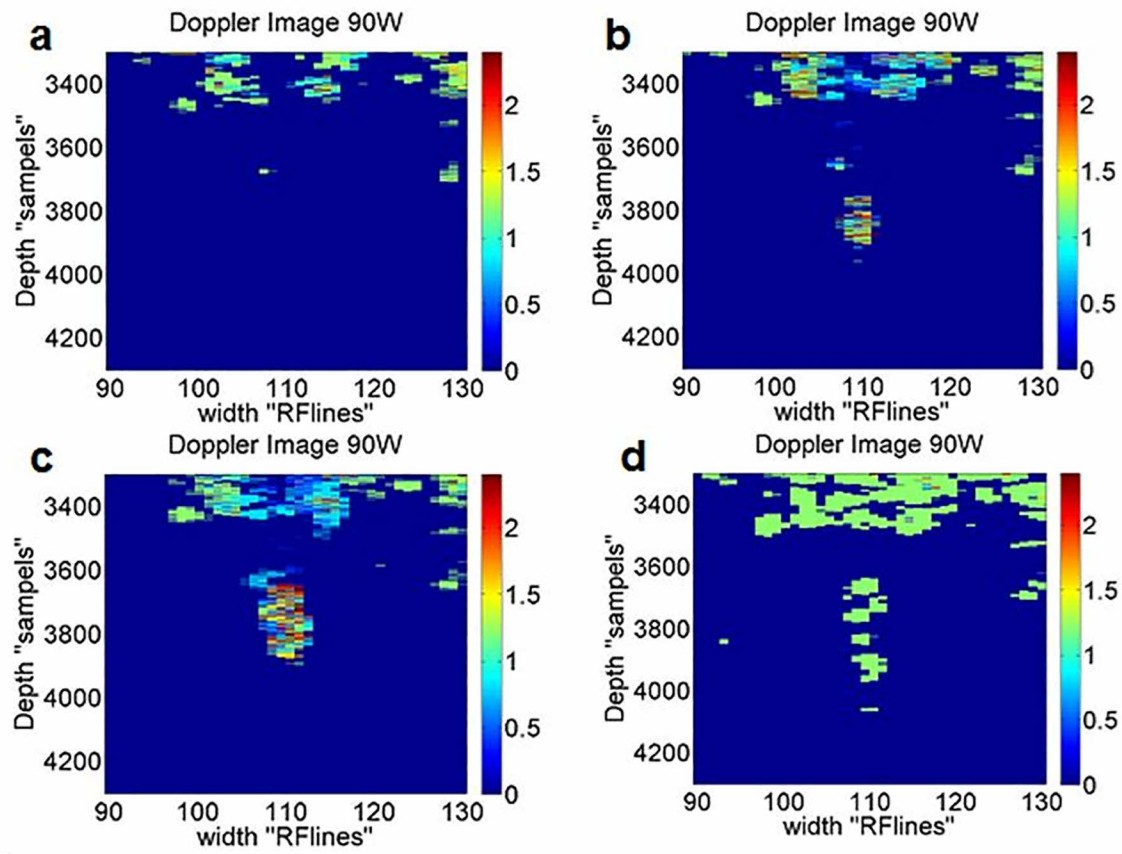

**Fig 5. Constructed Doppler images: (a), (b), and (c) during HIFU exposure, and (d) long after HIFU, corresponding to frames 13, 26, 39, and 67, respectively.** Colorbar indicates frequency shift (Hz).

To compare the lesion size in the constructed images with the measured lesion size from photographs provided in the dataset, we created Table 2 with the size data available for lesions under HIFU exposure at acoustic powers of 90, 110, and 130 W. In [19], researchers cut the tissue and took photographs. They included a ruler in their photographs to serve as a reference for length measurement. For instance, coagulated tissue from 110W HIFU exposure and its dimensions are illustrated in Fig 6.

In the original photographs, the samples were cut into two pieces, resulting in two close but not identical measurements for the dimensions of the ablated tissue. Due to the lack of access to the exact size of the lesion in the experimental study, and since we only have photographs of the cut tissue, we measured the depth and width in both right and left cuts and then used the average of these measurements. It can be seen that the lesion's width and depth monitored by the constructed images are in good match with the original size, with less than a 5% difference. This indicates that the constructed images provide a reliable representation of the actual lesion dimensions. Standard deviation (SD), confidence intervals (CI), and p-values were also calculated from 16 frames of constructed images taken after HIFU treatment. The low SD values, narrow 95% CI, and p-values show a strong agreement between the constructed images and the actual data. Overall, the method provides a reliable representation of lesion dimensions. The constructed images and photographs of ablated tissue with HIFU exposure at 90, 110, and 130 W are shown in Fig 7.

**Table 2. Comparison of lesion depth and width induced by HIFU exposure at 90, 110, and 130 W, showing actual values (from photographs), measured means (from constructed images), standard deviations (SD), percentage errors, 95% confidence intervals (CI), and p-values.**

| Power | Parameter | Actual Value | Mean (Measured) | SD | Percentage Error | 95% CI | p-value |
|-------|-----------|--------------|-----------------|------|------------------|------------|---------|
| 130 W | Depth (mm) | 14.4 | 14.45 | 0.13 | 0.35% | 14.45 ± 0.07 | 0.14 |
|       | Width (mm) | 12 | 12.01 | 0.19 | 0.08% | 12.01 ± 0.10 | 0.84 |
| 110 W | Depth (mm) | 14.28 | 14.46 | 0.14 | 1.26% | 14.46 ± 0.07 | 0.02 |
|       | Width (mm) | 8.9 | 8.64 | 0.2 | 2.92% | 8.64 ± 0.11 | 0.01 |
| 90 W  | Depth (mm) | 11.43 | 12 | 0.22 | 4.99% | 12.00 ± 0.12 | < 0.001 |
|       | Width (mm) | 7 | 7 | 0.2 | 0% | 7.00 ± 0.11 | 1 |

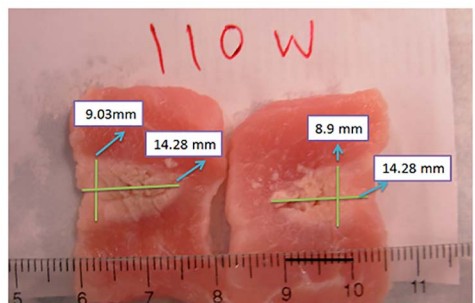

**Fig 6. Dimensions of the lesion in the captured photograph of tissue exposed to 110 W HIFU.**

## 4. Discussion

The potential of Doppler ultrasound imaging for monitoring HIFU-induced thermal lesions was examined in this study. As an alternative to MRI and other ultrasound-based techniques, we developed a real-time, cost-effective method that uses the Twinkling Artifact (TA) to detect and visualize lesions. Our approach shows benefits such as lower implementation complexity, transferability, and ease of integration into existing clinical systems. By focusing on frequency shifts, this method offers a novel tool for real-time thermal lesion monitoring. The results emphasize the viability of using TA as a marker for thermal lesions, opening the door for additional research and clinical translation.

In this work, we present a new method for using Doppler imaging to monitor HIFU-induced thermal lesions. Although the Twinkling Artifact (TA) for cavitation detection in HIFU therapy has been studied in the past by Khokhlova et al. [50] and Tong Li et al. [51], the use of TA for thermal lesion monitoring is still largely unexplored. Our work bridges this gap by examining the presence and characteristics of TA in the lesion area, offering a new tool for real-time assessment of HIFU-induced thermal damage. This is the first attempt to use TA for lesion monitoring, providing a practical and affordable alternative to current methods. By focusing on thermal lesions, our method complements existing approaches and improves the accuracy and safety of HIFU therapy.

The developed method offers significant advantages over MRI and other ultrasound-based techniques in terms of implementation complexity and cost. Unlike MRI, which requires expensive equipment and specialized facilities, our approach uses Doppler imaging systems that are widely available in clinical settings. This makes it a more accessible and cost-effective solution for real-time HIFU lesion monitoring. Additionally, compared to advanced ultrasound-based methods like Local Harmonic Imaging, Vibro-acoustography, and shear modulus imaging, our method eliminates the need for complex setups, such as high-frame-rate systems or additional transducers for tissue excitation. For instance, Arnal et al. [27] demonstrated that constructing shear wave images requires a high-frame-rate system (17,000 frames per second),

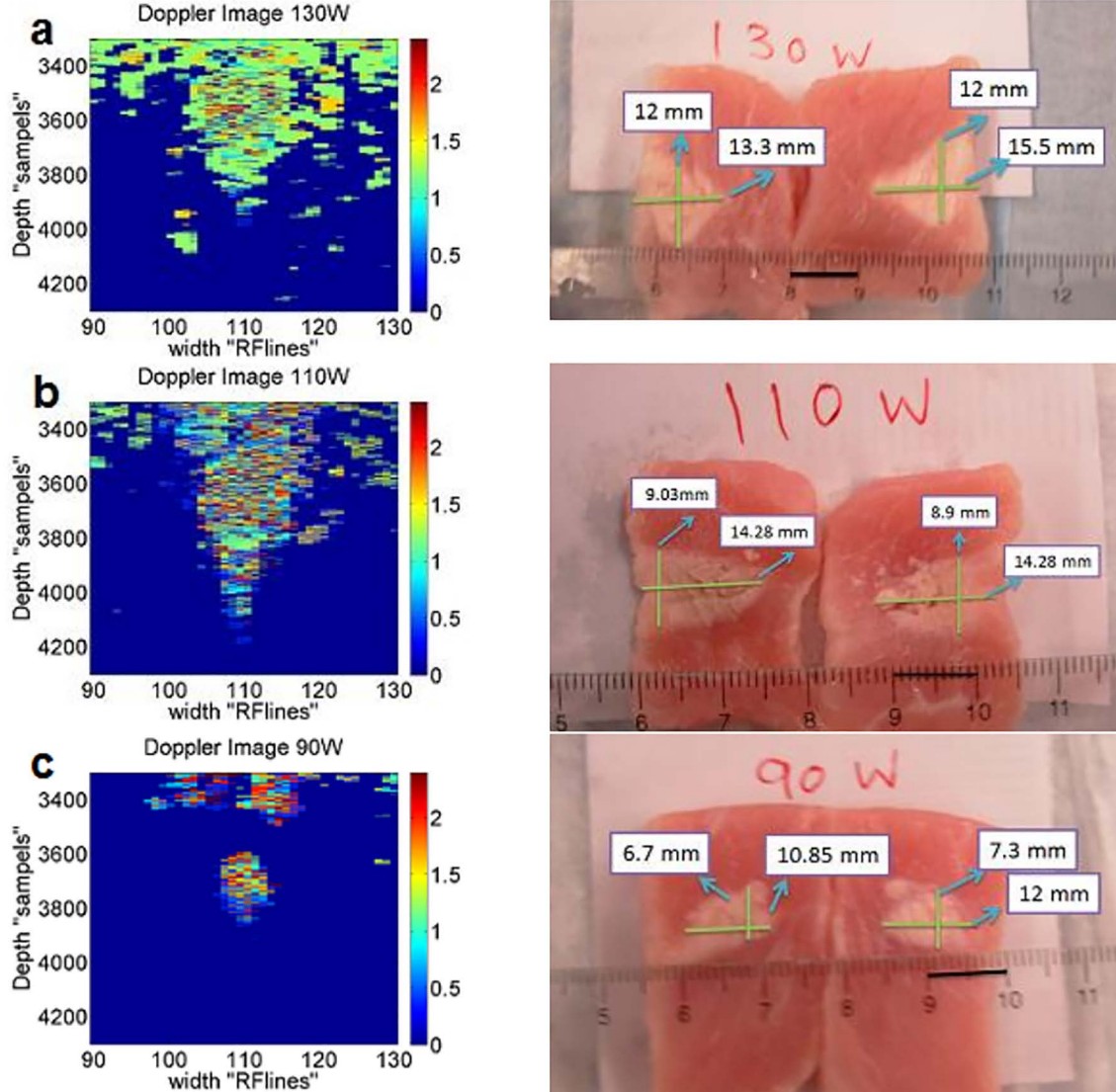

**Fig 7. (a), (b), and (c) are constructed images (Hz) and photographs of lesions with HIFU exposure at 130, 110, and 90 W, respectively.**

which is both costly and sensitive to physiological motions. In contrast, our method simplifies the monitoring process by relying solely on Doppler imaging, making it more practical for clinical use.

Ultrasound elastography also faces several limitations in guiding HIFU therapy. It relies on assumptions of linear elasticity, symmetrical tissue response, and homogeneous tissue properties, which are often challenged by the dynamic changes in tissue elasticity during HIFU treatment. Furthermore, in vivo applications are complicated by physiological movements and backscattered waves from bones, leading to inaccuracies in elastograms. Vibro-acoustography [24] and Local Harmonic Imaging [29] require additional transducers to generate acoustic radiation force and stimulate tissue oscillations, adding complexity to the treatment setup. In contrast, our proposed method does not require tissue excitation or high-frame-rate imaging systems. It utilizes standard Doppler imaging systems, which are already integrated into commercial ultrasound devices, making it easier to implement in clinical practice.

This method has two main limitations: the lack of in vivo validation and the challenges posed by physiological movements. While the ex vivo results are promising, they do not account for the complexities of living tissues, such as variations in tissue properties and patient movement. It is important to emphasize that the method is best suited for stationary tissues and should avoid areas with significant physiological movement, such as the heart and lungs. These limitations highlight the need for future in vivo trials to validate the method under realistic conditions and ensure its reliability in clinical settings.

To address these limitations and refine the method, future studies should focus on several key areas. First, using commercially available Doppler imaging systems, researchers can explore additional frequency features from RF signals to improve accuracy and reliability. Second, extracting the exact relationship between TA intensity, ultrasound frequency, and signal amplitude is essential for optimizing the method for clinical use. Factors influencing TA, such as acoustic radiation force (ARF), tissue depth, and tissue characteristics, should also be investigated. Additionally, future experimental setups should enable real-time monitoring of lesion area, boundaries, width, and depth, addressing the current reliance on post-treatment analysis. Merging this method with ultrasound-based thermometry, as demonstrated in Malekzadeh et al. [53] by the authors of this paper, could further enhance monitoring capabilities by combining thermal and structural information. Additionally, future studies exploring the influencing factors on the twinkling artifact could be valuable in validating this method for HIFU thermal therapy. Finally, the simultaneous use of MRI and Doppler ultrasound in future studies would enable a comprehensive quantitative error analysis. MRI can provide high-accuracy lesion dimensions and boundaries, making it an ideal dataset for validating ultrasound-based methods in lesion monitoring. This study represents the first step toward developing a reliable, clinically viable tool for HIFU lesion monitoring.

## 5. Conclusion

In this study, we investigated an ultrasound Doppler imaging approach for monitoring HIFU thermal therapy. This ultrasound-based method can be effective in expanding and developing the use of HIFU in therapeutic procedures. The findings from our study indicated that the twinkling artifact occurs significantly in Doppler ultrasound images of thermal lesions and that it can detect and monitor HIFU-induced lesions. Our results showed that Doppler imaging can monitor the formation of lesions during HIFU treatment and has a low error (or high sensitivity) in showing and measuring the dimensions of the lesions.

The results of this study showed that HIFU-induced lesions can be monitored during formation. Additionally, the lesions' size can be precisely measured by ultrasound Doppler imaging. These two factors can facilitate the HIFU therapy procedure and improve cancer treatment.

Compared to other ultrasound-based methods, the twinkling artifact in Doppler imaging offers simplicity, lower cost, and more precise localization of the thermal ablation within the tissues. These advantages of simplicity, lower cost, and efficacy make this method, which can suitable for integration into current ultrasonography systems, facilitating HIFU therapy's adoption and utilization in clinical procedures.

Our suggestions for future research in this field include validating this method in in vivo subjects, and applying machine learning-based image segmentation to estimate the volume of coagulated tissue. These recommendations would consider dynamic tissue properties and blood flow and develop an automatic approach for real-time HIFU exposure monitoring that could lead to higher precision and safety in ultrasound-guided thermal therapy.

## Author contributions

**Conceptualization:** Hamid Behnam.

**Data curation:** Jahangir Tavakkoli.

**Formal analysis:** Amirhossein Jamallivani.

**Investigation:** Amirhossein Jamallivani.

**Methodology:** Amirhossein Jamallivani, Hamid Behnam.

**Project administration:** Hamid Behnam.

**Software:** Amirhossein Jamallivani.

**Supervision:** Hamid Behnam.

**Writing – original draft:** Amirhossein Jamallivani.

**Writing – review & editing:** Hamid Behnam, Jahangir Tavakkoli.

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
