## [Decision Letter · Decision Letter 0]

3 Nov 2024

Dear Dr. Behnam,

We look forward to receiving your revised manuscript.

Kind regards,

Arka Bhowmik, Ph.D.

Academic Editor

PLOS ONE

Journal Requirements:

2. Check the last line of the abstract to ensure it is the same.  The reason for this check is to ensure that the AEs and Reviewers are sent correct information to allow them to make a good decision on whether they can manage/review the manuscript.  Only send back for a change if the abstract on EM and in the manuscript are VASTLY different.

Reviewers' comments:

Reviewer's Responses to Questions

**Comments to the Author**

1. Is the manuscript technically sound, and do the data support the conclusions?

Reviewer #1: Partly

Reviewer #2: Partly

2. Has the statistical analysis been performed appropriately and rigorously?

Reviewer #1: N/A

Reviewer #2: No

3. Have the authors made all data underlying the findings in their manuscript fully available?

Reviewer #1: No

Reviewer #2: No

4. Is the manuscript presented in an intelligible fashion and written in standard English?

Reviewer #1: Yes

Reviewer #2: No

Reviewer #1: The paper titled "Monitoring High-Intensity Focused Ultrasound (HIFU) Thermal Therapy by Ultrasound Doppler Imaging Using Twinkling Artifact" explores an approach for monitoring HIFU-induced thermal lesions in tissue by leveraging the twinkling artifact (TA) in Doppler ultrasound. The study utilizes real-time Doppler imaging to detect frequency shifts caused by HIFU exposure in ex-vivo porcine tissue, creating visual markers of lesion formation. The findings indicate that TA in Doppler images can accurately monitor lesion dimensions during HIFU ablation with less than 10% error, presenting this method as a potentially effective alternative to MRI-guided HIFU monitoring. The approach's simplicity, cost-efficiency, and compatibility with existing ultrasound systems are highlighted as significant advantages over MRI and other ultrasound methods. There are several major concerns regarding this study, along with suggestions that could improve the quality of the paper.

Major Concerns and Suggestions:

1. The authors frequently use the term “novel” throughout the manuscript. However, there are already studies available that explore the use of the twinkling artifact in Doppler imaging to monitor HIFU, some of which are mentioned below. The authors should clearly elaborate on the novelty of their work compared to previous studies and ensure that all relevant prior research is cited. Additionally, I suggest that they avoid using the word “novel” and remove it from all sections of the manuscript.

https://doi.org/10.1121/1.4800366

DOI: 10.1109/TUFFC.2014.006502

2. As mentioned above as well, the introduction should focus on the work that has already been done in this field. All the available references in this field, including the most recent ones from 2024 on HIFU and HIFU monitoring, should be carefully added.

3. The methodology section requires further detail to improve reproducibility. Key aspects, such as the specific Doppler signal processing steps, frequency thresholds, and image construction parameters, need more explicit descriptions.

4. The manuscript omits important machine-specific parameters, such as Doppler frequency and pulse repetition settings, which could impact reproducibility across different systems.

5. While the ex-vivo results are promising, there is no in-vivo validation. Including a discussion on the limitations of ex-vivo results and plans for future in-vivo trials is necessary to account for physiological variations.

6. While the paper briefly compares Doppler TA with MRI, a more comprehensive analysis contrasting the proposed method with existing ultrasound-based techniques (e.g., elastography, local harmonic imaging) would strengthen the discussion. This could help emphasize its unique advantages or address comparable limitations.

7. The paper overlooks some technical challenges, such as variability in TA intensity across different depths and tissue types, which could impact clinical implementation. Including a discussion on how these factors might influence TA consistency and proposing solutions would enhance the study’s practicality.

8. Beyond lesion size, additional quantitative assessments, such as boundary detection accuracy and frequency shift consistency, would help validate the method's precision. A quantitative error analysis of lesion boundary accuracy and TA intensity over time would further support the method's effectiveness and reliability.

9. Potential Doppler image artifacts, such as signal noise or low-intensity TA regions, are not addressed, which may impact lesion boundary detection.

10. The rationale behind selecting specific power levels (90W, 110W, 130W) is not detailed, potentially affecting reproducibility.

11. The study does not discuss how varying thermal doses impact TA visibility, potentially influencing lesion monitoring under different HIFU settings. The title of the paper explicitly mentions HIFU thermal therapy, implying that the thermal factor of HIFU would be the dominant factor in the therapy rather than the mechanical factor. However, there is no mention of the temperature increase amount due to each of the powers considered.

12. Potential clinical implementation challenges, such as patient movement and variability in tissue properties, are not discussed.

13. The study lacks a sensitivity analysis of Doppler TA detection, which would clarify the Doppler method’s performance across different lesion sizes and depths.

14. In the end, while the idea of monitoring HIFU by using the twinkling artifact of ultrasound Doppler imaging is a quite interesting topic, the presentation of results is not sufficiently scientific, leading to issues with result interpretation. I suggest improving the discussion section with a more quantitative analysis of the obtained results. Additionally, mentioning the drawbacks of this work and suggesting possible study directions to address these technical issues would be beneficial as well.

Reviewer #2: Review Comments for Manuscript PONE-D-24-41591:

Monitoring High-Intensity Focused Ultrasound Thermal Therapy by Ultrasound Doppler Imaging Using Twinkling Artifact

General Assessment:

This manuscript aims to introduce a novel approach for monitoring High-Intensity Focused Ultrasound (HIFU) thermal therapy using Doppler ultrasound imaging to detect twinkling artifacts (TAs) for lesion monitoring. The research is timely, given the increasing focus on non-invasive therapeutic modalities, and offers an interesting alternative to Magnetic Resonance Imaging (MRI) for HIFU lesion monitoring. However, several critical issues limit its suitability for publication in its current form. The manuscript would benefit from significant revisions to improve the clarity, rigor, and reproducibility of the presented method.

Major Comments:

Methodology and Experimental Design:

The manuscript lacks sufficient methodological details, particularly regarding the Doppler signal processing steps. Essential steps such as noise reduction techniques and parameter settings for Fast Fourier Transform (FFT) are vaguely described, which may hinder reproducibility.

The rationale for choosing specific parameters, such as sampling frequency and Doppler imaging intervals, is not clearly justified. Clarification is essential for verifying the robustness of the technique under different settings.

The study's experimental design relies on ex-vivo porcine tissue samples, which limit its clinical relevance. Extending the work to in-vivo studies is essential to validate the method for real-time HIFU lesion monitoring in a clinical context.

Data Analysis and Interpretation:

The authors report a less than 10% error in depth and width measurements for coagulated tissue dimensions; however, statistical validation is missing. Including statistical metrics, such as confidence intervals or p-values, would strengthen the claim.

The frequency analysis and Doppler image construction methods are introduced but not supported with quantitative analyses of accuracy or precision compared to established methods.

The constructed Doppler images (Figure 5) require more clarity, especially regarding the artifacts that are intended to represent TAs. The images should be annotated and analyzed with better-defined metrics.

Technical Limitations and Applicability:

Despite discussing the limitations of existing MRI and ultrasound-based methods, the manuscript does not provide a comprehensive comparison of how the proposed Doppler imaging approach surpasses or falls short of these alternatives. Explicit benchmarks or performance comparisons would be beneficial.

The proposed method’s sensitivity and specificity in detecting TAs related to HIFU lesions are not adequately evaluated, limiting the manuscript's practical value in guiding clinical applications.

Lack of Innovation and Literature Gap:

Although the use of Doppler imaging to monitor HIFU treatment is potentially valuable, the manuscript does not sufficiently differentiate itself from prior works on TAs in sonography. The novelty of the approach is undermined by a lack of unique insights or innovative techniques. A more thorough literature review that establishes a genuine gap is necessary.

Ethics and Data Availability:

There is no explicit ethical consideration or discussion on data accessibility. Given that the study could have implications for human subjects in future clinical applications, ethical considerations need to be addressed. Additionally, data availability should be transparent to ensure reproducibility.

Minor Comments:

The manuscript has several typographical and grammatical errors, which detract from the readability and professional presentation. A thorough proofreading is recommended.

Figures, particularly 5 and 6, lack sufficient annotations, which makes interpreting the results challenging. Including color-coded legends and scale bars would improve clarity.

The reference section includes several incomplete citations, with missing links to data sources and studies.

**Do you want your identity to be public for this peer review?** For information about this choice, including consent withdrawal, please see our Privacy Policy

Reviewer #1: No

Reviewer #2: No

---

## [Author Response · Author response to Decision Letter 1]

9 Feb 2025

All raised points are addressed in the response to reviewers file.

---

## [Decision Letter · Decision Letter 1]

21 Feb 2025

Dear Dr. Behnam,

Thank you for submitting your manuscript to PLOS ONE. After careful consideration, we feel that it has merit but does not fully meet PLOS ONE’s publication criteria as it currently stands. Therefore, we invite you to submit a revised version of the manuscript that addresses the points raised during the review process.

We look forward to receiving your revised manuscript.

Kind regards,

Kisoo Kim, Ph.D

Academic Editor

PLOS ONE

Journal Requirements:

Additional Editor Comments:

Dear Authors,

As Your manuscript, titled below, has been evaluated: "Monitoring High-Intensity Focused Ultrasound Thermal Therapy by Ultrasound Doppler Imaging Using Twinkling Artifact"

Some typographical errors and issues with English grammar have been identified. I kindly ask you to review and proofread the manuscript to improve its grammar and overall readability.

Reviewers' comments:

Reviewer's Responses to Questions

**Comments to the Author**

Reviewer #1: (No Response)

Reviewer #2: All comments have been addressed

2. Is the manuscript technically sound, and do the data support the conclusions?

Reviewer #1: Partly

Reviewer #2: Yes

3. Has the statistical analysis been performed appropriately and rigorously?

Reviewer #1: Yes

Reviewer #2: Yes

4. Have the authors made all data underlying the findings in their manuscript fully available?

Reviewer #1: Yes

Reviewer #2: Yes

5. Is the manuscript presented in an intelligible fashion and written in standard English?

Reviewer #1: No

Reviewer #2: Yes

Reviewer #1: I thank the authors for addressing the comments. Most of the concerns have been answered sufficiently, but some issues remain regarding the responses. Additionally, some technical refinements are necessary in the manuscript.

Unaddressed or Incompletely Addressed Points

1. Justification for Parameter Selection: The justification for choosing specific FFT parameters, noise thresholds, and segmentation window sizes is not detailed. The explanation of Doppler signal extraction and processing could be expanded. Specifically, the choice of a 10-sample window and the noise thresholding method require additional justification. Were these parameters optimized through testing or based on prior literature?

2. Quantitative Comparison of Doppler TA to Other Modalities: While the authors compared Doppler TA to MRI and ultrasound elastography, a quantitative comparison (e.g., sensitivity, specificity, accuracy) is lacking. The discussion on alternative imaging modalities (MRI, elastography, etc.) is useful, but quantitative performance comparisons (e.g., sensitivity/specificity, resolution) would strengthen the argument for Doppler TA’s clinical relevance.

3. Figure Modifications: Some figures (e.g., Figures 5 and 6) were updated based on reviewer comments but still need modification. The color bar explanation should not be only in the caption. The figure should be self-explanatory, allowing readers to grasp all relevant information at a glance. The color bar lacks a title indicating what the colors represent. Although mentioned in the caption, the figure itself should clearly display this information. Please add a title to the color bar in each of the figures.

4. Statistical Validation Details: Although standard deviation (SD) and confidence intervals were added, a more detailed explanation of the statistical methods used for lesion measurement validation is missing. Further details on how SD and confidence intervals were calculated would improve transparency. Additionally, were interobserver variability or repeatability tests performed?

Minor Technical Refinements

1. Terminology Consistency: There are inconsistencies in the use of terms such as "slow-time signals," "fast-time signals," and "RF signals." Ensure that terminology is uniform throughout the manuscript.

2. Equation Numbering and Formatting: Some equations (e.g., Doppler shift formula) are not properly referenced within the text. Ensure that equations are numbered correctly and referred to in explanations.

3. Language and Grammar Check: Although the authors claim that proofreading was done, it does not appear to have been performed by a native English speaker, as many minor grammatical errors remain. A final proofread for clarity and conciseness is essential. Some examples of errors are as follows:

• Missing Articles ("a," "an," "the")

Example 1: "The Twinkling Artifact (TA) is color Doppler artifact caused by acoustic radiation force and consequent tissue vibration during Doppler imaging."

Correction: "The Twinkling Artifact (TA) is a color Doppler artifact caused by the acoustic radiation force and the consequent tissue vibration during Doppler imaging."

Issue: "A" was missing before "color Doppler artifact," and definite articles ("the") were needed.

Example 2: "Figure 5 and 6 could benefit from clearer annotations."

Correction: "Figures 5 and 6 could benefit from clearer annotations."

Issue: When referring to multiple figures, "Figures" (plural) should be used instead of "Figure."

• Subject-Verb Agreement

Example 1: "The Doppler images that was constructed show the lesion formation."

Correction: "The Doppler images that were constructed show the lesion formation."

Issue: "Images" is plural, so the verb should be "were" instead of "was."

Example 2: "These results suggest that Twinkling Artifact is useful in monitoring thermal lesions and are in agreement with previous studies."

Correction: "These results suggest that the Twinkling Artifact is useful in monitoring thermal lesions and is in agreement with previous studies."

Issue: "Twinkling Artifact" is singular, so it should be "is" instead of "are."

• Inconsistent Tense Usage

Example 1: "In the study, we show that the Doppler signal extraction method provides reliable results and confirmed the presence of TA."

Correction: "In this study, we show that the Doppler signal extraction method provides reliable results and confirm the presence of TA."

Issue: The sentence shifts between present ("show") and past ("confirmed")—both should be in present tense for consistency.

Example 2: "The proposed method was tested and demonstrates a high level of accuracy."

Correction: "The proposed method was tested and demonstrated a high level of accuracy."

Issue: "Was tested" is past tense, so "demonstrates" should also be changed to past tense ("demonstrated") for consistency.

• Clarity and Redundancy

Example 1: "This study aims to investigate and analyze the role of the Twinkling Artifact for the purpose of lesion monitoring."

Correction: "This study analyzes the role of the Twinkling Artifact in lesion monitoring."

Issue: "Investigate and analyze" are redundant, and "for the purpose of" can be replaced with a simpler "in."

Example 2: "The obtained results are consistent and in agreement with previous studies."

Correction: "The results are consistent with previous studies."

Issue: "Consistent" and "in agreement" mean the same thing—one should be removed.

There are many many more issues regarding awkward phrasing and wordiness. Reading this manuscript is difficult due to the overwhelming number of grammatical errors. A simple grammar checker, such as Grammarly, would eliminate most of these issues!!!

Reviewer #2: (No Response)

**Do you want your identity to be public for this peer review?** For information about this choice, including consent withdrawal, please see our Privacy Policy

Reviewer #1: No

Reviewer #2: No

---

## [Author Response · Author response to Decision Letter 2]

13 Apr 2025

Please see the response to reviewers file.

---

## [Editor Report · Decision Letter 2]

2 May 2025

Monitoring High-Intensity Focused Ultrasound Thermal Therapy by Ultrasound Doppler Imaging Using Twinkling Artifact

PONE-D-24-41591R2

Dear Dr. Behnam,

We’re pleased to inform you that your manuscript has been judged scientifically suitable for publication and will be formally accepted for publication once it meets all outstanding technical requirements.

Kind regards,

Kisoo Kim, Ph.D

Academic Editor

PLOS ONE
---

## [Editor Report · Acceptance letter]

PONE-D-24-41591R2

PLOS ONE

Dear Dr. Behnam,

I'm pleased to inform you that your manuscript has been deemed suitable for publication in PLOS ONE. Congratulations! Your manuscript is now being handed over to our production team.

Kind regards,

on behalf of

Dr. Kisoo Kim

Academic Editor

PLOS ONE